# Rhodium-catalyzed selective direct arylation of phosphines with aryl bromides

Dingyi Wang[1,4], Mingjie Li[1,4], Chengdong Shuang[2✉], Yong Liang [1], Yue Zhao [1], Minyan Wang [1✉] & Zhuangzhi Shi [1,3✉]

The widespread use of phosphine ligand libraries is frequently hampered by the challenges associated with their modular preparation. Here, we report a protocol that appends arenes to arylphosphines to access a series of biaryl monophosphines via rhodium-catalyzed P(III)-directed *ortho* C–H activation, enabling unprecedented one-fold, two-fold, and three-fold direct arylation. Our experimental and theoretical findings reveal a mechanism involving oxidative addition of aryl bromides to the Rh catalyst, further *ortho* C–H metalation via *a* four-membered cyclometalated ring. Given the ready availability of substrates, our approach opens the door to developing more general methods for the construction of phosphine ligands.

[1] State Key Laboratory of Coordination Chemistry, Chemistry and Biomedicine Innovation Center (ChemBIC), School of Chemistry and Chemical Engineering, Nanjing University, Nanjing 210093, China. [2] State Key Laboratory of Pollution Control and Resources Reuse, School of the Environment, Nanjing University, Nanjing 210093, China. [3] School of Chemistry and Chemical Engineering, Henan Normal University, Xinxiang, Henan 453007, China. [4]These authors contributed equally: Dingyi Wang, Mingjie Li. ✉email: shuangchendong@nju.edu.cn;wangmy@nju.edu.cn; shiz@nju.edu.cn

Phosphines have found numerous applications in all facets of chemical science[1–5]. Among them, biaryl monophosphines have emerged as a class of privileged ligands for transition metals in a variety of transformations, especially in cross-coupling reactions[6–11]. To date, multiple generations of biaryl monophosphines have been designed by Buchwald[12,13] and other groups[14–17], and many of them are now commercialized (Fig. 1a). Traditionally, these compounds can be produced via a one-pot protocol through the addition of an aryl metal reagent to an in situ-generated benzyne, then phosphination of intermediate with a chlorophosphine reagent enabled by a copper catalyst (Fig. 1b)[18]. This approach is efficient, but its use in sensitive organometallic species is often limited, requiring pre-installation of halides into the substrates and complicated operating steps. Compared to traditional coupling methods, the direct arylation strategy through C–H activation has emerged as a valuable methodology that enables the formation of biaryl compounds with excellent atom and step economy[19–23]. We hypothesized that the catalytic arylation of C–H bond in phosphines might allow an alternative, but a far easier path to biaryl monophosphines.

Typically, the assistance of metal-coordinating directing groups in substrates can lead to excellent regioselectivity[24–30]. Within this paradigm, transition metal-catalyzed C–H arylation of phosphine oxides through O-chelation has been developed to build biaryl monophosphines[31–35]. However, this method requires additional steps for preinstallation and removal of O atoms. Due to the strong coordination between transition metals and P(III) atoms, arylphosphines have long been known to form four-membered chelate rings via ortho C–H metalation, but catalytic variants represent a critical challenge[36]. Until recently,

we and Takaya group have respectively reported catalytic ortho C–H borylation[37,38] and silylation[39] of arylphosphines. However, a further palladium-catalyzed Suzuki–Miyaura or Hiyama cross-coupling with aryl halides needs to be used for construction of biaryl monophosphines. Therefore, direct arylation of arylphosphines to access diverse biaryl monophosphines in a catalytic fashion is still in high demand.

Here, we report a rhodium-catalyzed P(III)-directed C–H activation of arylphosphines with aryl halides to rapidly access a library of biaryl monophosphines. Notably, onefold, twofold, and threefold C–H activation can proceed by steric control of aryl bromides, providing biaryl monophosphine ligands with sterically encumbered architectures and electronically tuned substituents in a tunable way (Fig. 1c). In the first C–H arylation, the reaction proceeds via a four-membered chelate ring of arylphosphines, and the second and third arylation of the in situ formed biaryl phosphines via a six-membered chelate ring[40–43]. Therefore, the use of sterically hindered aryl bromides only can show one-fold C–H activation; Treatment of aryl bromides with moderate size can undergo two-fold C–H activation; The selection of aryl bromides only with para-substituents can lead to three-fold C–H activation.

## Results

**Reaction design.** Our investigation began with the direct arylation of PPh₃ (**1a**) using a sterically hindered aryl bromide **2a** (Table 1). Treatment of [Rh(cod)Cl]₂ (2.5 mol%) as the catalyst along with K₂CO₃ (2.0 equiv) as the base in THF at 130 °C for 24 h allowed us to get the desired coupling product **3aa** in 21% yield through onefold C–H activation (entry 1). The solvent effect was then evaluated, and 1,4-dioxane proved to be the best (entries 2-3).

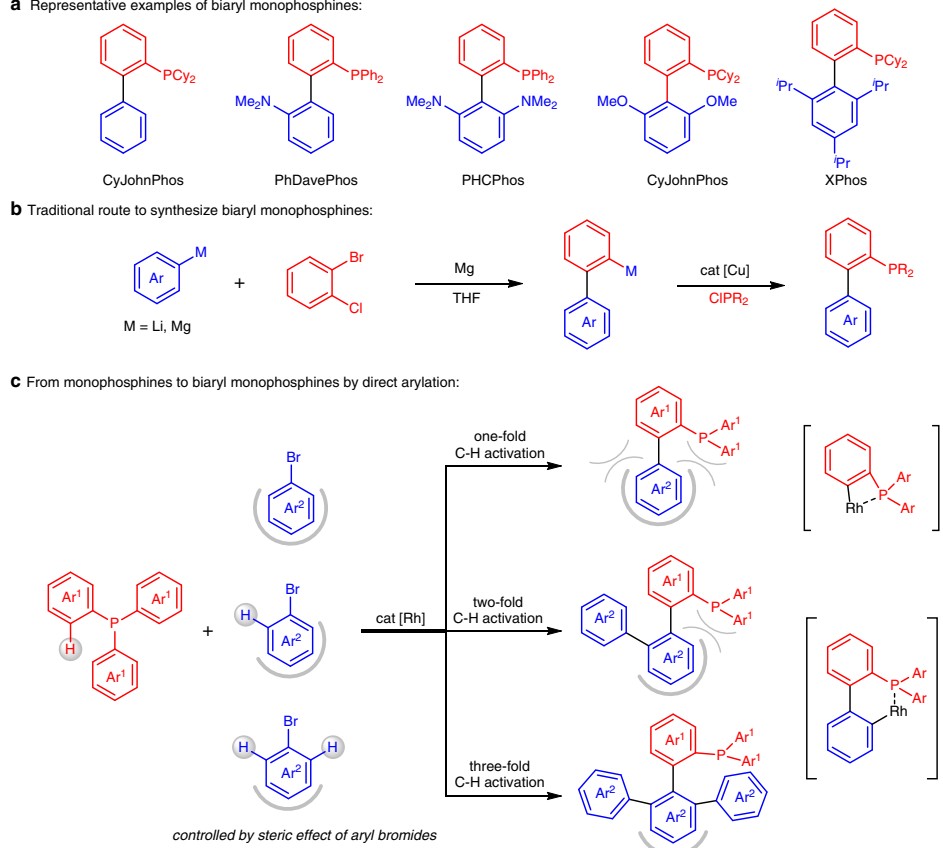

**Fig. 1 Construction of biaryl monophosphine ligands. a** Some commercially available biaryl monophosphines. **b** Palladium-catalyzed carbon-phosphorus bond metathesis. **c** Rhodium-catalyzed tunable direct arylation of phosphines with aryl bromides.

**Table 1 Reaction development[a].**

| Entry | X | cat [TM] (mol%) | Base (equiv) | Solvent | Yield of 3aa (%)[b] |
|---|---|---|---|---|---|
| 1 | Br (**2a**) | [Rh(cod)Cl]₂ (2.5) | K₂CO₃ (2.0) | THF | 21 |
| 2 | Br (**2a**) | [Rh(cod)Cl]₂ (2.5) | K₂CO₃ (2.0) | DME | 35 |
| 3 | Br (**2a**) | [Rh(cod)Cl]₂ (2.5) | K₂CO₃ (2.0) | 1,4-dioxane | 42 |
| 4 | Br (**2a**) | [Rh(cod)Cl]₂ (2.5) | K₂CO₃ (5.0) | 1,4-dioxane | 68 |
| 5 | Br (**2a**) | [Rh(cod)Cl]₂ (2.5) | K₂CO₃ (5.0) | 1,4-dioxane[c] | 77 (72)[d] |
| 6 | Br (**2a**) | [Rh(cod)Cl]₂ (2.5) | Na₂CO₃ (5.0) | 1,4-dioxane[c] | 56 |
| 7 | Br (**2a**) | [Rh(cod)Cl]₂ (2.5) | Li₂CO₃ (5.0) | 1,4-dioxane[c] | 15 |
| 8 | Br (**2a**) | [Rh(cod)Cl]₂ (2.5) | LiO^tBu (5.0) | 1,4-dioxane[c] | 74 |
| 9[e] | Br (**2a**) | [Rh(cod)Cl]₂ (2.5) | K₂CO₃ (5.0) | 1,4-dioxane[c] | 65 |
| 10 | Br (**2a**) | [Rh(coe)₂Cl]₂ (2.5) | K₂CO₃ (5.0) | 1,4-dioxane[c] | 76 |
| 11 | Br (**2a**) | [Rh(OAc)₂]₂ (2.5) | K₂CO₃ (5.0) | 1,4-dioxane[c] | 27 |
| 12 | Br (**2a**) | [Ir(cod)Cl]₂ (2.5) | K₂CO₃ (5.0) | 1,4-dioxane[c] | 0 |
| 13 | Br (**2a**) | Pd(OAc)₂ (5.0) | K₂CO₃ (5.0) | 1,4-dioxane[c] | 0 |
| 14 | I (**2a'**) | [Rh(cod)Cl]₂ (2.5) | K₂CO₃ (5.0) | 1,4-dioxane | 62 |

[a]Reaction conditions: cat [TM] (2.5–5.0 mol %), **1a** (0.2 mmol), **2a** (0.6 mmol), base (2.0–5.0 equiv) in solvent (1.0 mL) at 130 °C for 24 h, under argon.
[b]Determined by GC.
[c]Reducing solvent to 0.3 mL.
[d]Isolated yield after chromatography.
[e]Reaction at 110 °C.

Further increasing the stoichiometric amount of base to 5.0 equiv is critical to the high conversion of the reaction (entry 4). It should be noted that increasing the concentration of reaction components can enhance the reactivity, affording product **3aa** in 77% yield (entry 5). Other bases such as Na₂CO₃ and Li₂CO₃ showed much lower results for this reaction (entries 6–7), and the use of LiO^tBu also maintained good reactivity leading to compound **3aa** in 74% yield (entry 8). Under these reaction conditions, conducting the reaction at 110 °C led to a lower conversion (entry 9). Other rhodium sources, such as [Rh(coe)₂Cl]₂ and [Rh(OAc)₂]₂, were also effective for this transformation, albeit with lower yields (entries 10–11). However, other transition metals, such as [Ir(cod) Cl]₂ and Pd(OAc)₂, failed to trigger this transformation (entries 12–13). In addition to aryl bromide, aryl iodide **2a'** could be coupled with PPh₃ (**1a**), affording the product **3aa** in 62% yield (entry 14).

**Scope of the methodology.** With optimized reaction conditions in hand, we first explored the scope of one-fold direct arylation between tertiary phosphines **1** and arylbromides **2** (Fig. 2). Using PPh₃ (**1a**) as the coupling partner, arylbromides **2b**-**c** bearing sterically hindered benzhydryl and silyl ether groups at the *meta* positions formed desired products **3ab**-**ac** with good efficiency. Less sterically hindered arylbromides **2d**-**e** and bromonaphthalene **2f** were also compatible with excellent chemoselectivity, and twofold C–H activation could be inhibited by shortening the reaction time and lowering the reaction temperature. Importantly, the selection of OMe at *ortho* position of arylbromides had a strong impact on the reactivity, other functional groups such as NMe₂, SMe, and Ac led to very low conversions under the current reaction conditions. In addition, the reaction is also amenable to 9-bromophenanthrene (**2g**) and 3,5-di-substituted arylbromides **2h**-**i**, affording desired products **3ag**-**ai** in 60–75% yields. Subsequently, a variety of triarylphosphines were investigated with arylbromide **2a**. Triarylphosphines are generally competent, with

a diverse range of electron-neutral (**3ba**-**bc**), electron-donating (**3da**-**3ea**), and electron-withdrawing (**3fa**-**3ga**) groups.

We next sought to evaluate the scope of the two-fold C–H activation by steric control of aromatic halides (Fig. 3). We were pleased to find that two molecules of 2-bromoanisole (**1j**) could participate in this reaction through tandem C–H activation to generate product **3aj** in 77% yield under slightly modified reaction conditions. To determine the structure, a crystal of compound **3aj** was generated and subjected to X-ray crystallographic analysis. Bromoanisole analogs **1k** and **1l** with F and Cl at the *para* position also generated desired products **3ak** and **3al** in 72% and 61% yields, respectively. Gratifyingly, substrate **1m** with a fluorene motif can be utilized in this transformation as well, generating product **3am** in 62% yield together with a monoarylation byproduct in trace amounts.

We further turned our attention to direct arylation of phosphines through threefold C–H activation (Fig. 4). Treatment of the simple bromoarene **2n**, PPh₃ (**1a**) could be arylated with three molecules of **2n**, providing compound **3an** in 66% yield. It's noted that the use of ^tBuOLi showed much higher reactivity than that of K₂CO₃ in these reaction conditions. This approach has a good substrate scope and is tolerant of a range of substituents in para positions of the aromatic ring. A broad range of bromoarenes bearing Me (**1o**), OMe (**1p**), F (**1q**), Cl (**1r**), and CF₃ (**1s**) substituents underwent threefold direct arylation and gave the corresponding products **3ao**-**at** in 51-73% yield. In addition, bromoferrocene (**1t**) was successfully introduced into the phosphine framework, affording the compound **3at** with three ferrocene motifs. The structures of products **3ap** and **3at** were respectively confirmed by X-ray analysis.

We also sought to explore the cascade C–H arylation of arylphosphines with two different aryl bromides. However, a mixture of biaryl phosphines was generated, because the excess amount of aryl halides needs to be employed. To solve this issue, we conducted the first direct arylation with PPh₃ (**1a**) and aryl

**Fig. 2 Substrate scope of one-fold direct arylation.** Reaction conditions: [Rh(cod)Cl]$_2$ (2.5 mol %), **1** (0.2 mmol), **2** (0.6 mmol), K$_2$CO$_3$ (1.0 mmol) in 1,4-dioxane (0.3 mL) at 130 °C for 24 h under argon. [a]Using [Rh(cod)Cl]$_2$ (5.0 mol%). [b]Reaction at 120 °C for 12 h. [c]Using [Rh(cod)$_2$]OTf (10.0 mol%).

**Fig. 3 Substrate scope of two-fold direct arylation.** Reaction conditions:[Rh(coe)$_2$Cl]$_2$ (5.0 mol %), **1a** (0.2 mmol), **2j–m** (1.0 mmol), K$_2$CO$_3$ (1.0 mmol) in 1,4-dioxane (0.5 mL) at 150 °C for 24 h under argon. [a]Yield of monoarylation product.

bromide **2f** in the presence of rhodium catalyst. The desired product **3af** was isolated in 45% yield, which could be arylated with another aryl bromide **2o** through C–H activation to provide biaryl phosphine **3afo** efficiently (Fig. 5).

## Discussion

To gain insight into the reaction pathway, some mechanistic experiments were performed (Fig. 6). The reaction of [Rh(cod)

Cl]$_2$ with PPh$_3$ (**1a**) in 1,4-dioxane formed the well-known Wilkinson's catalyst, which could further react with aryl bromide 2a to give product 3aa in 67% yield (Fig. 6a). This result indicated that the complex was a visible intermediate in this catalytic reaction. In addition, a kinetic isotope effect (KIE) of 2.4 was observed from five parallel reactions of 1a/d-1a with aryl bromide 2a, suggesting that the C–H cleavage step was rate-determining (Fig. 6b)[44].

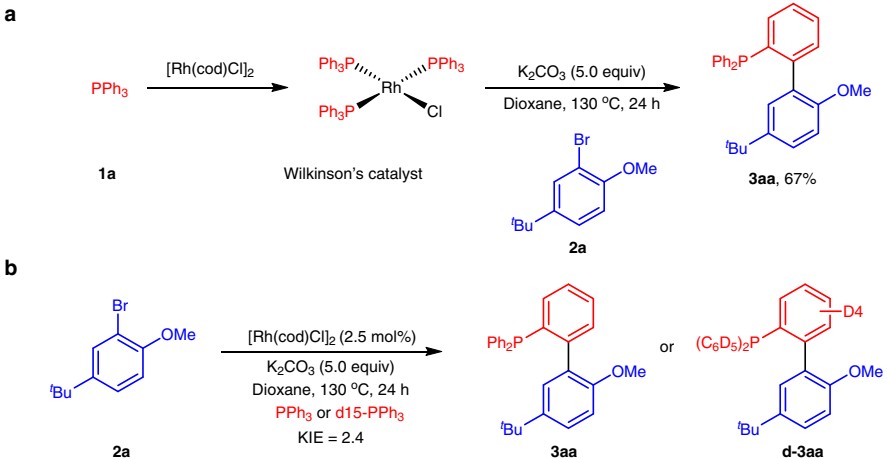

**Fig. 4 Substrate scope of threefold direct arylation.** Reaction conditions: [Rh(coe)₂Cl]₂ (5.0 mol %), **1a** (0.2 mmol), **2n-t** (1.0 mmol), LiO$^t$Bu (1.2 mmol) in 1,4-dioxane (0.5 mL) at 150 °C for 24 h under argon.

**Fig. 5 Twofold direct arylation of PPh₃ (1a) with two different aryl bromides.** Reaction conditions: **a** [Rh(cod)Cl]₂ (2.5 mol %), **1a** (0.2 mmol), **2f** (0.6 mmol), K₂CO₃ (1.0 mmol) in 1,4-dioxane (0.3 mL) at 120 °C for 12 h, under argon. **b** [Rh(coe)₂Cl]₂ (5.0 mol %), **3af** (0.1 mmol), **2o** (0.3 mmol), K₂CO₃ (0.5 mmol) in 1,4-dioxane (0.25 mL) at 150 °C for 36 h, under argon.

**Fig. 6 Mechanistic experiments. a** Investigation of Wilkinson's catalyst as a key intermediate. **b** KIE experiments of **1a** and d15-**1a**.

To better understand the selective C–H direct arylation process, density functional theory (DFT) calculations[45–48] were conducted with model substrates **1a** and **2a** (Fig. 7). Initially, the dimer Rh catalyst is associated with **1a** to form Wilkinson's catalyst, which then undergoes ligand exchange to generate four-coordinate complex **INT1A**. Bromoarene **2a** coordinates with the Rh center to form intermediate **INT2A** through an energy barrier of

31.4 kcal mol⁻¹, which facilitates subsequent C–Br bond cleavage (black line, Fig. 7a). *Ortho*-C–H metalation is a competitive pathway for the oxidative addition of the C–Br bond (blue line, Fig. 7a). The *ortho*-C–H bond undergoes oxidative addition to the Rh(I) center through transition state **TS2B** with a free energy of 41.7 kcal·mol⁻¹, which is 9.3 kcal mol⁻¹ higher than transition state **TS3A** (41.7 kcal mol⁻¹ vs 32.4 kcal mol⁻¹). These

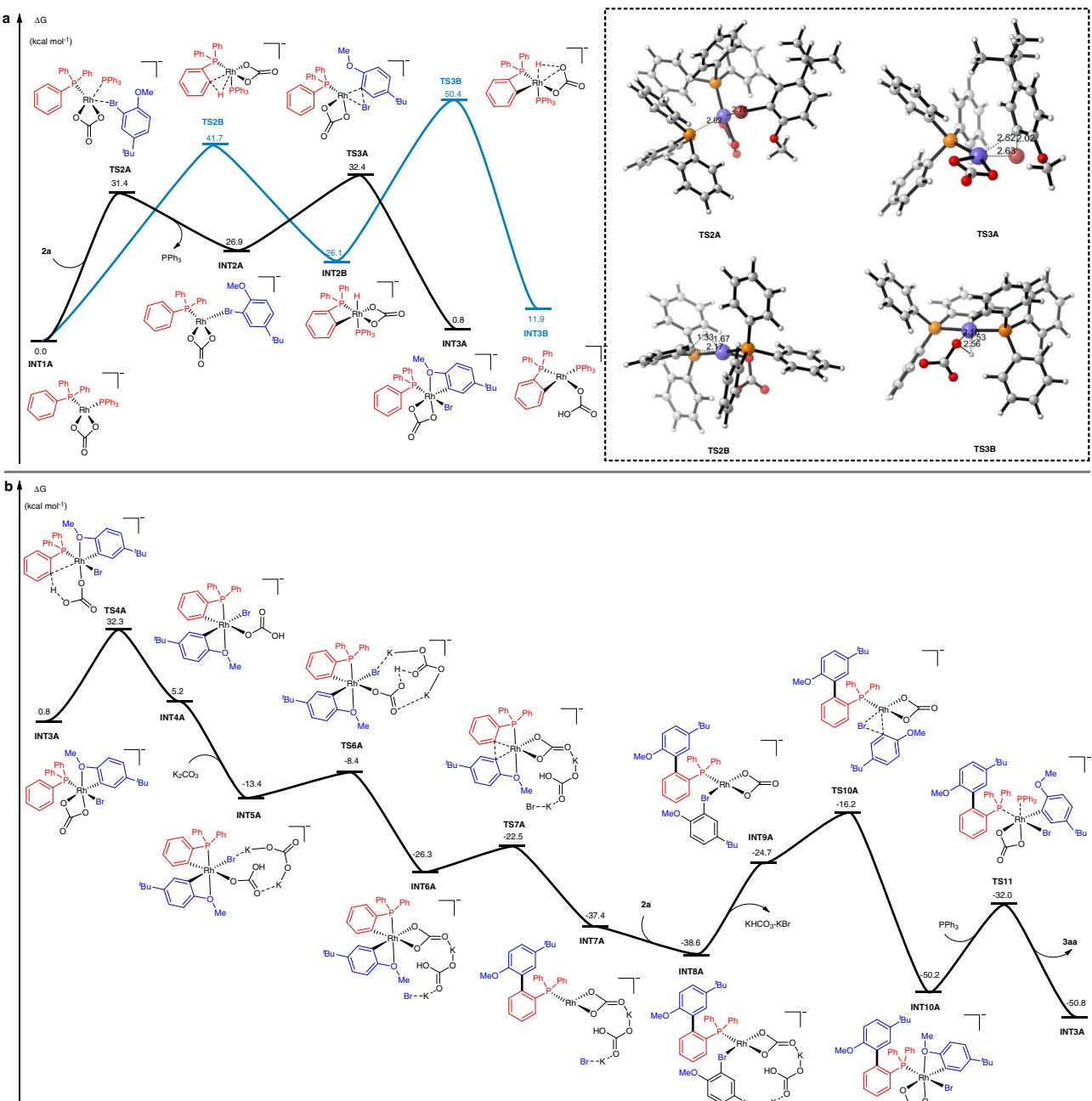

**Fig. 7 Plausible reaction mechanism for onefold direct arylation between substrates 1a and 2a. a** DFT-computed free energies of the two competitive pathways for C–Br and C–H bond oxidative addition. **b** The free energy profiles for the catalytic cycle of the onefold direct arylation. Energies are in kcal mol$^{-1}$ and bond lengths are in Å. The energies were computed at the M06/SDD-6-311 + G(d, p)/SMD (solvent = 1,4-dioxane) level of theory, with geometries optimized at the B3LYP/SDD–6-31 G(d) level.

calculated results indicate that the most favorable pathway involves the oxidative addition of the C–Br bond to the rhodium(I) center, resulting in intermediate **INT3A**. Then, **INT3A** occurs *ortho*-C–H metalation to form **INT5A** through a concerted metalation deprotonation (CMD) process[49–52] with a 31.5 kcal mol$^{-1}$ energy barrier (Fig. 7b). This process possesses very similar high energy to the C–Br bond oxidative addition (32.3 kcal mol$^{-1}$ vs. 32.4 kcal mol$^{-1}$). Furthermore, the calculation for a deuterated transition state of **TS4A** was also performed (see Supporting Information). The energy barrier of C–D metalation is 0.9 kcal mol$^{-1}$ higher than that of C–H, in consistent with the experimental KIE results, illustrating that C–H metalation is involved in the rate-determining step. $K_2CO_3$ as a base was

found to be indispensable in this transformation, which abstracts a Br atom and a proton from complex **INT4A** in a concerted manner via transition state **TS6A** with an energy barrier of only 5.0 kcal mol$^{-1}$. Followed by reductive elimination, the aryl group connected with Rh(III) center transfers to the *ortho*-position of phosphines to deliver intermediate **INT7A**. The excess amount of **2a** then coordinates at the vacant site of **INT7A** to generate intermediate **INT8A**. Finally, the C–Br bond oxidative addition and further ligand exchanges occur to result in the formation of the desired product **3aa** and the regeneration of catalyst **INT3A** to complete the catalytic cycle.

Subsequently, DFT calculations were conducted to investigate the chemoselectivity of twofold arylation (Fig. 8). Using

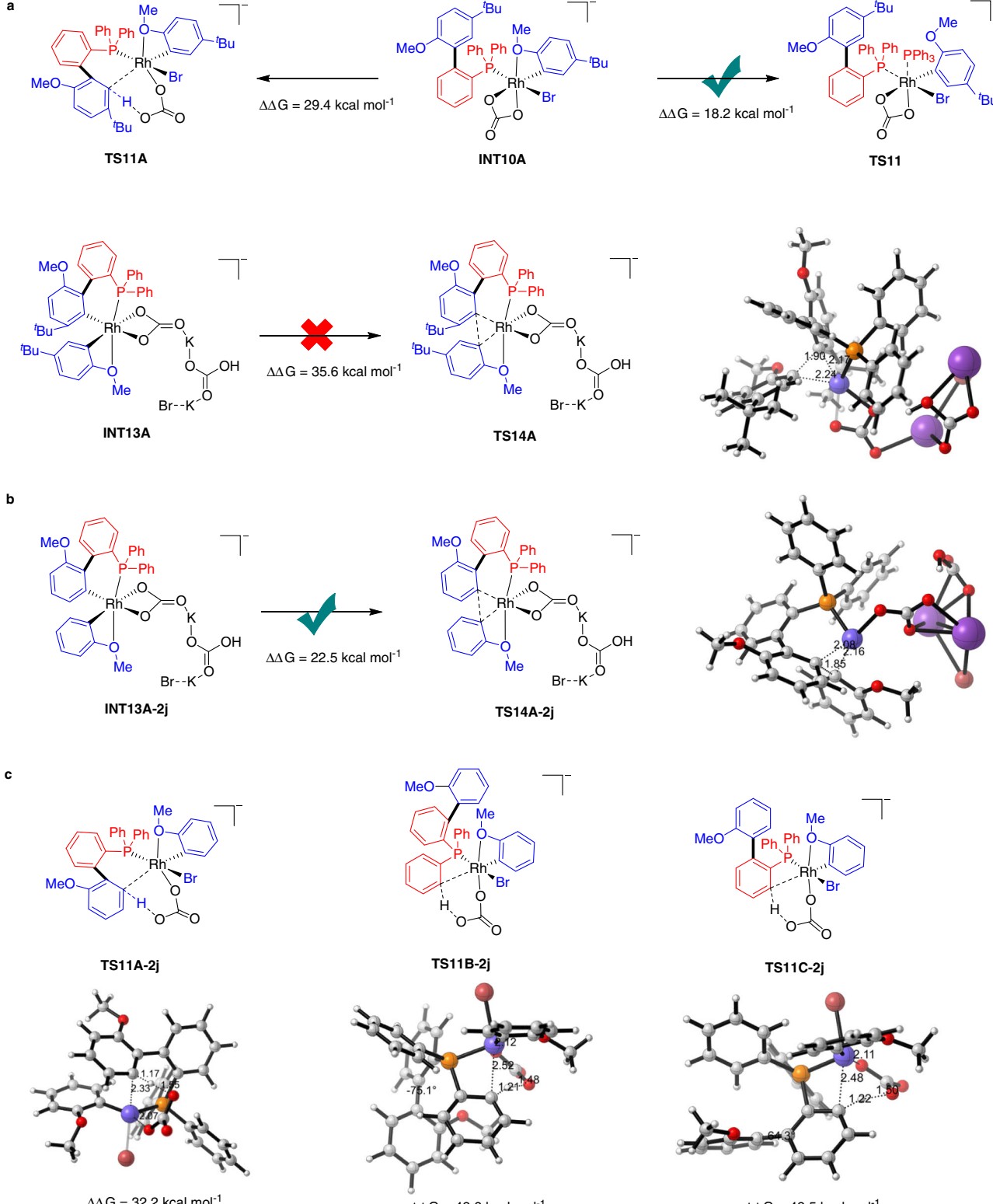

**Fig. 8 Mechanistic experiments. a** the calculated key transition states and intermediates for the twofold direct arylation of PPh₃ and **2a**. **b** the calculated key transition states and intermediates for the twofold direct arylation of PPh₃ and **2j**. **c** DFT-computed free energies for the three competitive C-H metalation pathways in the second-fold direct arylation of PPh₃ with **2j**. Energies are in kcal mol⁻¹ and bond lengths are in Å.

bromoarene **2a** with sterically hindered substituents at the *meta* positions as the substrate, ligand exchange of **INT10A** with **1a** is found to be a favorable process, and its barrier height is 11.2 kcal mol⁻¹ lower than the C-H metalation through six-

membered cyclic transition state **TS11A** (18.2 kcal mol⁻¹ vs 29.4 kcal mol⁻¹), suggesting that onefold arylation will proceed to the end until **1a** was completely consumed out (Fig. 8a). In addition, the *meta*-substitutions of bromoarenes increase the

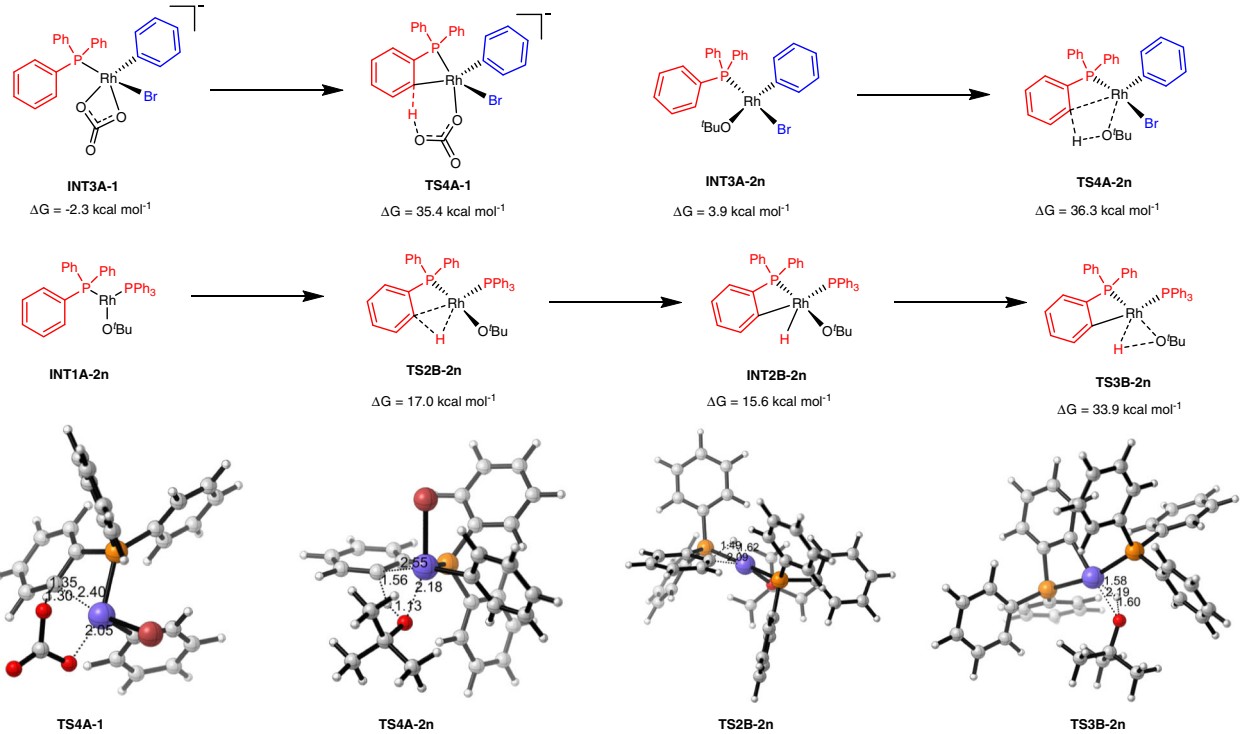

**Fig. 9 The effect of $^{t}$BuOLi in the threefold direct arylation.** Energies are in kcal mol$^{-1}$ and bond lengths are in Å.

difficulty of the second reductive elimination, making the activation barrier of transition state **TS14A** as high as 35.6 kcal mol$^{-1}$. Therefore, controlling the reaction time and increasing the steric hindrance of *meta*-substituents could effectively tune the chemoselectivity. Instead, using bromoarene **2j** as the substrate, the reductive elimination of intermediate **INT13A-2j** proceeds smoothly via transition state **TS14A-2j** with an energy barrier of 22.5 kcal mol$^{-1}$, thus producing the twofold arylation product **3aj**, in accordance with the experimental results (Fig. 8b)[40]. The second arylation on different sites was also calculated (Fig. 8c). The *ortho'*-C-H activation undergoes a CMD process through six-membered cyclic transition state **TS11A-2j** with a free energy of 32.2 kcal mol$^{-1}$, which is much lower than that of four-membered cyclic transition state **TS11B-2j** (42.8 kcal mol$^{-1}$) and **TS11C-2j** (42.5 kcal mol$^{-1}$). In the disfavored transition states **TS11B-2j** and **TS11C-2j**, the dihedral angle of biphenyl is deformed to 75.1° and 64.3°, respectively, indicating the distinct repulsion between the incorporated aryl group and second aryl group of P, thus increasing the activation energy barriers to >40 kcal/mol.

The use of $^{t}$BuOLi as the base in the threefold C-H arylation was studied as well (Fig. 9). The above-used K$_2$CO$_3$ showed lower reactivity than $^{t}$BuOLi in this transformation. In the onefold arylation process, the *ortho*-substituents of bromoarene donate electron effect into the vacant d orbital of Rh, which facilitates the dissociation of oxygen connected with Rh and increases the alkalinity of the carbonate, thus making the energy barrier of the CMD process lower to 31.5 kcal mol$^{-1}$. However, the bromoarene without *ortho*-substituents are suitable substrates in the threefold arylation process, the carbonate-assisted CMD process via **TS4A-1** has an energy barrier of 37.7 kcal mol$^{-1}$, which is much higher than the corresponding process in onefold C-H arylation (37.7 kcal mol$^{-1}$ vs 31.5 kcal mol$^{-1}$). For the LiO$^{t}$Bu used in the experiment, anion-assisted CMD transition state **TS4A-2n** is calculated to be 36.3 kcal mol$^{-1}$ in energy relative to **INT1A-2n**, indicating that the

mechanism of alkoxy base assisted CMD process is also difficult. To explore a more plausible reaction mechanism, the C-H bond oxidative addition with Rh(I) through transition state **TS2B-2n** was also calculated. This step is more favorable, since the free energy of rate-determining step is 33.9 kcal·mol$^{-1}$ relative to **INT1A-2n**, which is 1.5 kcal·mol$^{-1}$ higher than the overall barrier of 32.4 kcal mol$^{-1}$ in onefold arylation. This explains well that the threefold arylation is able to proceed well when the reaction temperature was elevated from 130 to 150 °C. The whole energy profiles for the threefold C-H arylation have been shown in Supporting Information.

In summary, we have developed an effective method for the direct arylation of arylphosphines with aryl bromides enabled by rhodium catalysts. The reaction can proceed through onefold, twofold, and threefold C-H activation by steric control of aromatic halides, affording a series of biarylphosphine ligands with architecture and electronically tuned substituents. Mechanistic experiments and density functional theory calculations showed the preferred pathway for this tunable direct arylation process. Further applications of the developed phosphine ligand library, as well as other C-H functionalization of arylphosphines, are underway.

## Methods

**General procedures for synthesis of 3**. To an oven-dried Schlenk tube, arylphosphines **1** (1.0 equiv, 0.20 mmol), arylbromides **2** (3.0 equiv, 0.60 mmol), [Rh(cod)Cl]$_2$ (2.5 mol%, 2.5 mg, 0.005 mmol), K$_2$CO$_3$ (5.0 equiv, 138 mg, 1.0 mmol) were dissolved in 1,4-dioxane (0.3 mL). The mixture was stirred at 130 °C under argon for 24 h. Upon the completion of the reaction, the solvent was removed. The crude mixture was directly subjected to column chromatography on silica gel using petroleum-ether/ EtOAc as eluent to give the desired products **3**.

## Data availability

The crystallography data have been deposited at the Cambridge Crystallographic Data Center (CCDC) under accession number CCDC: 2082986 (**3aj**), 2082987 (**3ap**), 2082989 (**3at**) and can be obtained free of charge from www.ccdc.cam.ac.uk/getstructures.

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

## Acknowledgements

We would like to thank the National Natural Science Foundation of China (Grants 22025104, 22171134, 21972064, and 21901111), the Fundamental Research Funds for the Central Universities (Grant 020514380254) for their financial support, and the "Innovation & Entrepreneurship Talents Plan" of Jiangsu Province for their financial support. We are also grateful to the High-Performance Computing Center (HPCC) of Nanjing University for doing the numerical calculations in this paper on its blade cluster system. The project was also supported by Open Research Fund of School of Chemistry and Chemical Engineering, Henan Normal University.

## Author contributions

Z.S. conceived and designed the study, and wrote the paper. D.W. and M. L. performed the experiments, mechanistic studies, and analyzed the data. C.S. guided the study and made contributions during the revision. M.W. and Y. L. performed the DFT calculations. Y.Z. performed the crystallographic studies.

## Competing interests

The authors declare no competing interests.
