## [Peer Review File · Nature Communications]

REVIEWER COMMENTS

Reviewer #1 (Remarks to the Author):

The manuscript by Shi and co-workers reported their investigation on Rh(I)-catalyzed C–H bond arylation of triarylphosphines with aryl bromides to form biarylphosphines (BAPs). Generally, BAPs are prepared via C–P bond formation. Here, C–H bond arylation allows the preparation of unique structures and gives rise to an extensive library of ligands quickly. They control the mono-, di-, and tri-arylation by playing on the phosphines/ArBr ratio and temperature. The selectivity seems to depend on the nature of the aryl bromides. A substituent at the meta position favors the monoarylation. Ortho and para-substituted aryl bromides give biaryl products, while para-substituted afforded triarylated products. However, there are no general rules. DFT calculations suggest that the reaction proceeds via CMD mechanism rather than OA. It showed why meta-substituted ArBr give monoarylation. The work was carried out with high integrity; this transformation was never reported; this transformation allows access to chiral phosphines. For all these reasons, I strongly recommend its publication after answering the following questions:

1. The authors should explain why the second arylation occurred at the ortho'-position and not on a second Ar group of the P? Did they observe this product?
2. Is it possible to arylate the three aryl groups of $P(Ar)_3$ using a larger amount of ArBr.
3. The stoichiometry for the ArBr should be more clearly indicated.
4. What is the reactivity/selectivity of ArI in this reaction?
5. Did the authors have tried to perform the two-fold (or three-fold) arylation using 2 or 3 different aryl bromides (or a mix between ArI and ArBr)?
6. The authors should explain why the substitution patterns of ArBr are so crucial for selectivity and end up with a comprehensive rule.

Reviewer #2 (Remarks to the Author):

Wang and Shi reported a new protocol for the synthesis of biaryl monophosphines via a Rh-catalyzed, P(III)-directed ortho C–H arylation reaction of arylphosphines with aryl halides. In particular, the reaction underwent sterically-controlled one-fold, two-fold, and three-fold arylation, enabling direct access to a variety of biarylphosphines possessing aryl substituents at 2'- and 6'-positions. A plausible reaction mechanism was proposed based on a KIE experiment and DFT calculations, clarifying the origin of the chemoselectivity and effect of the base. Development of metal-catalyzed direct C–H bond transformations of P(III) compounds is one of rapidly growing areas in synthetic and organometallic chemistry, and among them, the ortho C–H arylation of arylphosphines has remained unachieved. This paper achieved it for the first time and demonstrated a certain level of novelty on this point, but in my opinion, not sufficient impact and novelty for publication in Nature Communications. This is largely based on the several precedents for the ortho C–H bond arylation and silylation reactions of arylphosphines developed by this group and others (ref 37-39). In addition, there are numerous examples of Rh- and Ru-catalyzed C–H arylation and alkylation reactions of biaryl phosphines with aryl- and alkyl halides, giving various 2'- and 6'-substituted biaryl phosphines reported by the author and others (e.g. ref 50). Therefore, I have to feel that this is a kind of extension work of these precedents applying the system to the current combination. Moreover, the chemoselectivity is largely substrate-dependent, limiting its synthetic utility. In other words, the monoarylation was achieved only with rather specific, sterically bulky aryl halides whereas the over-reaction easily proceeded with standard, less sterically demanding substrates. Although the author claimed the importance of biaryl phosphines, the synthesis of highly valuable Buchwald's biaryphosphines such as XPhos etc. was not demonstrated. The obtained biaryl phosphines in this paper are rather specific, and their synthetic utility are totally unclear. Consequently, I conclude that this work does not meet the very high bar of novelty and impact required for publication in Nature Communications. More specialized journals to synthetic organic chemistry would be suited.

Comments

- 1) Two-fold arylation products seem to contain substantial amounts of byproducts in their NMR spectra in SI. The author commented as "affording only a trace amount of one-fold direct activation product (<5%)" in line 83. Please provide spectral information of the one-fold arylation product and give the yield and selectivity accurately.
- 2) As commented above, there are many examples of Rh- and Ru-catalyzed C–H arylation and alkylation reactions of biaryl phosphines with aryl- and alkyl halides other than ref 50. These should be cited.
- 3) The author have reported a Rh-catalyzed C–H arylation reaction of biaryl phosphines (ref 50), which is closely related to the two-fold and three-fold arylation in this work. This should be

described properly in the main text, and more detailed investigations on the difference of reactions conditions, reactivity, and chemoselectivity should be carried out.

Reviewer #3 (Remarks to the Author):

This manuscript describes an interesting C-H arylation directed by phosphines. The authors highlight the ability to select aryl halides with particular steric hindrance to allow selectivity for mono-arylation, di-arylation, or tri-arylation. The initial findings of the manuscript are of great interest and could be made appropriate for this journal, but in the current form, the manuscript does not address important reactions that should have been examined and includes experimental results that are superfluous. Furthermore, the authors do not do a thorough analysis that explains apparently conflicting data between the experimental and computational results. This manuscript will require major revisions and additional experiments to warrant publication in Nature Communications.

Introduction

I find the connection between Morandi's work and this work to be unclear. Why is this critical to the current work. This should either be articulated better or excluded from the figure and a different point of emphasis made.

Results

Table 1: It is unclear to me why the authors did not examine alkoxide bases. It is clear that these bases are very good for triarylation. Do they work for monoarylation? Why not include these in this screen and determine their ability compared to the other bases. Leaves questions instead of answers later in the manuscript.

Figure 2: The only ortho-substituted aryl bromide used in this substrate scope is methoxy. Does nothing else work? If not, what happens? Does it need to be an alkoxy? Can it be alkyl? What is the role of this group? A much more representative substrate scope should be included here.

Figure 3: The authors chose to only use one aryl halide for the sequential reactions. Is it possible to use two different aryl halides by doing a monoarylation, followed by another aryl bromide? I think the answer is no, but the authors should address this limitation.

It is not obvious why there is not a second arylation at the ortho C-H bond on the other side of the phosphorus atom. If you look closely at the computations, the energetics suggest that the two C-H insertion reactions are very similar in energy. The authors do not mention if any of this arylation product is formed. Why would it be disfavored? I think this casts a shadow on the quality of the computations if there is high selectivity and the computations put the energy barrier as equivalent. The authors should address in the text if any of the bis-ortho arylation is observed.

Figure 4: It is unclear to me why the authors included the CD data in this paper. It is a complete non-sequitur. There are no asymmetric reactions that are included and the assignment of the enantiomers is outside of the scope of the manuscript. At most, it could be noted that the two enantiomers of 3aj could be separated by HPLC.

Figure 6b: The aryl halide is mislabeled as 1a. This is rather confusing. I also found the indication of the deuterated substrates as unclear. PPh₃ should be labeled as -d₁₅ in the figure. Furthermore, the indication of d₁₄ on d-3aa is unclear. Each Ph group should be labeled as C₆D₅ and D₄ could be shown on the aryl group that is drawn out.

Figure 7: This figure would be much clearer if you replaced 1a with PPh₃. This is a simple fix that would increase clarity significantly.

Line 140: The text says that the calculated TS for TS5A is 31.5. Figure 7 says it is 32.3. Which one is correct? I would not expect a KIE for this reaction if it is not the rate determining step. It is very close to TS4A if the figure is correct, but not if the text is correct. Some comment on the subtle nature of the computations vs. the experiment would be helpful. I also think that a calculation for a deuterated substrate would be ideal to examine the difference.

Figure 8: The authors chose to do the reductive elimination TS for the t-Bu substrate and it is over the RDS, but not by a huge amount. I don't think this explains a product like 3ad. I suspect that RE step will be lower than the turnover limiting step. In this case it would be difficult to stop the reaction at 1 arylation if the computations are correct. The authors should show this calculation and comment on the size of that group and how it influences the selectivity. This can be seen in the lower temp and shorter reaction time for 3ad, but no comment is made.

Also in figure 8, it is notable that TS10A and TS10-2j are so similar in energy. Even more notable that TS10a is lower in energy than TS10a-2j, which is less congested. What does this tell you? Some comment on this would be ideal.

Supporting Information:

The SI is thorough and quite good, but the authors should include the melting points of the solid products.

Reviewer #4 (Remarks to the Author):

This work is concerned with a methodology development, reporting a Rh(I)-catalysed protocol that appends arenes to arylphosphines to access a series of biaryl monophosphines via P(III)-directed ortho C–H activation. Experimentally, the results are important and interesting, and well described. Overall, I believe that this work deserves publication in Nature Communications.

However, the computational results were rather poorly presented and discussed.

1. In most of the chemdraw structures in the potential energy profiles, the drawings do not reflect the coordination chemistry around the Rh metal center, making readers hard to comprehend the results. We all know that Rh(I) often adopts square planar geometry if it is 4-coordinate and Rh(III) adopts an octahedral if it is 6-coordinate. In all of the drawing, this kind of coordination feature is completely missing.
2. A potential energy profile for a catalytic cycle should show clearly the “starting” point (Active Catalyst + Reactants) and the “ending” point (The same Active Catalyst + Product). However, readers cannot see this in all the profiles presented in the main text and in the SI.
3. In the potential energy profiles presented, the ligand exchanges (e.g., INT2A + 2a  INT3a + 1a; INT6a to INT7a; and INT7a + 2a  INT3A + 3aa) are expected to have barriers. The authors do not calculate/include these, which could be high and very important.
4. Most species are mono-anionic. The authors do not show this in the chemdraw structures presented.
5. From Fig 7, the Ar-Br OA transition state and the concerted metalation deprotonation (CMD) transition state show almost the same stability. The data in fact show that the OA TS is lying slightly higher than the CMD TS, suggesting that the OA TS is very likely the rate-determining TS. How could the authors claim that the DFT results are consistent with the KIE experimental observation?
6. In Fig 7, TS4B seems to miss one phosphine ligand.

7. It is hard to understand how TS5B transforms to INT5B. This seems to jump too fast, because one more phosphine ligand is included. Then, from INT5B to INT6B, no transition states are involved?
8. Did the authors attempt to locate a direct CMD process that converts INT2A to a species containing HOAc as a ligand?
9. In Fig 4b, the theoretical ECDs are noticeably different from the experimental ECDs. I do not understand the claim "...were found to match well with the experimental ECD spectra ..."
10. The presentation and discussion on the two- and three-fold arylation reactions are very hard to comprehend. Rewriting of the part is necessary.

Reviewer #1

The manuscript by Shi and co-workers reported their investigation on Rh(i)-catalyzed C–H bond arylation of triarylphosphines with aryl bromides to form biarylphosphines (BAPs). Generally, BAPs are prepared via C–P bond formation. Here, C–H bond arylation allows the preparation of unique structures and gives rise to an extensive library of ligands quickly. They control the mono-, di-, and tri-arylation by playing on the phosphines/ArBr ratio and temperature. The selectivity seems to depend on the nature of the aryl bromides. A substituent at the meta position favors the monoarylation. Ortho and para-substituted aryl bromides give biaryl products, while para-substituted afforded triarylated products. However, there are no general rules. DFT calculations suggest that the reaction proceeds via CMD mechanism rather than OA. It showed why meta-substituted ArBr give monoarylation. The work was carried out with high integrity; this transformation was never reported; this transformation allows access to chiral phosphines. For all these reasons, I strongly recommend its publication after answering the following questions:

Response: Thanks for such positive comments.

1. The authors should explain why the second arylation occurred at the ortho'-position and not on a second Ar group of the P? Did they observe this product?

Response: Thanks for your questions. The second arylation on a second Ar group of P was not observed, since this process proceeds *via* a four-membered chelate ring, which possesses greater ring tension than that process *via* six-membered chelate ring. DFT calculation was also performed to calculate the energy barrier of these transformations. The ortho'-C-H activation undergoes a concerted metalation-deprotonation (CMD) process through six-membered cyclic transition state **TS11A-2j** with a free energy of $32.2 \text{ kcal} \cdot \text{mol}^{-1}$, which is much lower than that of four-membered cyclic transition state **TS11B-2j** ($42.8 \text{ kcal} \cdot \text{mol}^{-1}$) and **TS11C-2j** ($42.5 \text{ kcal} \cdot \text{mol}^{-1}$). This part was added in Fig 8c during the revision.

2. Is it possible to arylate the three aryl groups if $\text{P}(\text{Ar})_3$ using a larger amount of ArBr.

Response: When the reaction was carried out using a larger amount of ArBr (30 equiv.), we did not observe the mentioned products.

3. The stoichiometry for the ArBr should be more clearly indicated.

Response: We clearly indicated the stoichiometry in Fig 2-4 according to your suggestion.

4. What is the reactivity/selectivity of ArI in this reaction?

Response: Thanks for your suggestion. We added an example of ArI (Table 1, entry 14), which showed a slightly lower yield.

5. Did the authors have tried to perform the two-fold (or three-fold) arylation using 2 or 3 different aryl bromides (or a mix between ArI and ArBr)?

Response: Thanks for the constructive suggestion. We added a related example in Fig. 5.

6. The authors should explain why the substitution patterns of ArBr are so crucial for selectivity and end up with a comprehensive rule.

Response: Thanks for this constructive suggestion. According to your suggestion, we added a sketch map in Fig. 1c and the comprehensive rule in the text for the readers to understand. The selectivity for one-fold, two-fold, and three-fold C–H activation can proceed by steric control of aryl bromides. In the first C–H arylation, the reaction proceeds via a four-membered chelate ring of arylphosphines, and the second and third arylation of the in-situ formed biaryl phosphines via a six-membered chelate ring. Therefore, the use of sterically hindered aryl bromides only can show one-fold C–H activation; Treatment of aryl bromides with moderate size can undergo two-fold C–H activation; The selection of aryl bromides only with para-substituents can lead to three-fold C–H activation.

Reviewer #2

Wang and Shi reported a new protocol for the synthesis of biaryl monophosphines via a Rh-catalyzed, P(III)-directed ortho C–H arylation reaction of arylphosphines with aryl halides. In particular, the reaction underwent sterically-controlled one-fold, two-fold, and three-fold arylation, enabling direct access to a variety of biarylphosphines possessing aryl substituents at 2'- and 6'-positions. A plausible reaction mechanism was proposed based on a KIE experiment and DFT calculations, clarifying the origin of the chemoselectivity and effect of the base. Development of metal-catalyzed direct C–H bond transformations of P(III) compounds is one of rapidly growing areas in synthetic and organometallic chemistry, and among them, the ortho C–H arylation of arylphosphines has remained unachieved. This paper achieved it for the first time and demonstrated a certain level of novelty on this point, but in my opinion, not sufficient impact and novelty for publication in Nature Communications. This is largely based on the several precedents for the ortho C–H bond borylation and silylation reactions of arylphosphines developed by this group and others (ref 37-39). In addition, there are numerous examples of Rh- and Ru-catalyzed C–H arylation and alkylation reactions of biaryl phosphines with aryl- and alkyl halides, giving various 2'- and 6'-substituted biaryl phosphines reported by the author and others (e.g. ref 50). Therefore, I have to feel that this is a kind of extension work of these precedents applying the system to the current combination. Moreover, the chemoselectivity is largely substrate-dependent, limiting its synthetic utility. In other words, the monoarylation was achieved only with rather specific, sterically bulky aryl halides whereas the over-reaction easily proceeded with standard, less sterically demanding substrates. Although the author claimed the importance of biaryl

phosphines, the synthesis of highly valuable Buchwald's biarylphosphines such as XPhos etc. was not demonstrated. The obtained biaryl phosphines in this paper are rather specific, and their synthetic utility are totally unclear. Consequently, I conclude that this work does not meet the very high bar of novelty and impact required for publication in Nature Communications. More specialized journals to synthetic organic chemistry would be suited.

Response: Thanks for your constructive comments, which are very helpful for us to improve the quality of this work. Here, I would like to demonstrate the novelty of this chemistry in details.

Functionalization of unactivated C–H single bonds is an efficient strategy for rapid generation of complex molecules from simpler ones. However, it is difficult to achieve selectivity when multiple inequivalent C–H bonds are present in the target molecule. The usual approach is to use δ -chelating directing groups, which lead to *ortho*-selectivity through a five/six-membered cyclometallated ring. Recently, template-directed the activation of distal meta-C–H bonds via more than ten-membered cyclometallated ring (distal distance) have also been developed. On the contrary, the catalytic C–H bond activation mode that proceeds through a four-membered cyclometallated ring (proximal distance) represents a critical challenge.

Transition-metal-mediated metallation of an aromatic C–H bond that is adjacent to a tertiary phosphine group in arylphosphines via a four-membered chelate ring was first discovered in 1968. Since 2019, our group has developed catalytic *ortho* C–H functionalization of arylphosphines, including C–H borylation (Angew. Chem. Int. Ed. 2019, 58, 2078) and silylation (Angew. Chem. Int. Ed. 2020, 59, 10909.). However, a further palladium-catalyzed Suzuki–Miyaura or Hiyama cross-coupling with aryl halides need to be used for construction of biaryl monophosphines. Therefore, direct arylation of arylphosphines to access diverse biaryl monophosphines in a catalytic fashion is still in high demand. Therefore, direct arylation of arylphosphines to biaryl monophosphines in a catalytic fashion represent a critical challenge, which is still in high demand.

Here, we develop a simple and general approach to synthesize the biaryl monophosphines through rhodium-catalysed selective direct arylation of phosphines and aryl bromides with excellent atom and step economy. Unexpectedly, one-fold, two-fold, and three-fold C–H activation can proceed by steric control of aromatic halides, providing phosphine ligands bearing sterically encumbered architectures and electronically tuned substituents in a tunable way. Our experimental and theoretical findings reveal a mechanism involving oxidative addition of aryl bromides to the Rh catalyst, further *ortho* C–H metalation via a four-membered cyclometalated ring. Given the ready availability of substrates, our approach opens the door to developing more general methods for the construction of phosphine ligands.

Therefore, I strongly hope you can reconsider the significance our work in light of these points. To further better understand the novelty of this work, we rewrote the introduction part during the revision.

Comments:

1) Two-fold arylation products seem to contain substantial amounts of byproducts in their NMR spectra in SI. The author commented as “affording only a trace amount of one-fold direct activation product (<5%)” in line 83. Please provide spectral information of the one-fold arylation product and give the yield and selectivity accurately.

Response: Thanks for this suggestion. During the revision, we double-checked all case in Figure 3 by ^{31}P NMR and GC-MS. The reaction of **2m** can afford only a trace amount of one-fold direct

activation product (8%). Because this byproduct can't be separated by column chromatography on silica gel, we can't provide the related spectral information. Alternatively, we gave the yield and selectivity accurately in Fig.3.

2) As commented above, there are many examples of Rh- and Ru-catalyzed C–H arylation and alkylation reactions of biaryl phosphines with aryl- and alkyl halides other than ref 50. These should be cited.

Response: Thanks for this kind suggestion. We added the related papers of Rh- and Ru-catalyzed C–H arylation and alkylation reactions of biaryl phosphines were added in refs 40-43.

3) The author have reported a Rh-catalyzed C–H arylation reaction of biaryl phosphines (ref 50), which is closely related to the two-fold and three-fold arylation in this work. This should be described properly in the main text, and more detailed investigations on the difference of reactions conditions, reactivity, and chemoselectivity should be carried out.

Response: Thanks for this kind suggestion. We added a related description in the introduction part as “In the first C–H arylation, the reaction proceeds via a four-membered chelate ring of arylphosphines, and the second and third arylation of the in-situ formed biaryl phosphines via a six-membered chelate ring.⁴⁰⁻⁴³”. It's important to consider that a chemical transformation depends not only on the products, but only on the starting materials. Here, Rh(I)-catalysed protocol that appends arenes to arylphosphines to access a series of biaryl monophosphines via P(III)-directed ortho C–H activation, enabling unprecedented one-fold, two-fold, and three-fold direct arylation.

Reviewer #3 (Remarks to the Author):

This manuscript describes an interesting C-H arylation directed by phosphines. The authors highlight the ability to select aryl halides with particular steric hindrance to allow selectivity for mono-arylation, di-arylation, or tri-arylation. The initial findings of the manuscript are of great interest and could be made appropriate for this journal, but in the current form, the manuscript does not address important reactions that should have been examined and includes experimental results that are superfluous. Furthermore, the authors do not do a thorough analysis that explains apparently conflicting data between the experimental and computational results. This manuscript will require major revisions and additional experiments to warrant publication in Nature Communications.

Introduction

I find the connection between Morandi's work and this work to be unclear. Why is this critical to the current work. This should either be articulated better or excluded from the figure and a different point of emphasis made.

Response: We changed the introduction according to your suggestion.

Results

Table 1: It is unclear to me why the authors did not examine alkoxide bases. It is clear that these bases are very good for triarylation. Do they work for monoarylation? Why not include these in this screen and determine their ability compared to the other bases. Leaves questions instead of answers later in the manuscript.

Response: Thanks for these questions. The use of alkoxide bases such as LiOtBu showed a

slightly lower yield for monoarylation, however treatment of K_2CO_3 for triarylation led to a dramatically lower conversion. We added this information in entry 8, Table 1.

Figure 2: The only ortho-substituted aryl bromide used in this substrate scope is methoxy. Does nothing else work? If not, what happens? Does it need to be an alkoxy? Can it be alkyl? What is the role of this group? A much more representative substrate scope should be included here.

Response: Thanks for the questions. We tried the ortho-substituted aryl bromides bearing methyl and F, which didn't work at the current reaction conditions. The methoxy group donates lone pairs into the vacant d orbital of Rh, which facilitates the dissociation of oxygen connected with Rh and increasing the alkalinity of the carbonate, thus lowering the activation energy of the CMD process to promote the reaction.

Figure 3: The authors chose to only use one aryl halide for the sequential reactions. Is it possible to use two different aryl halides by doing a monoarylation, followed by another aryl bromide? I think the answer is no, but the authors should address this limitation.

Response: Thanks for the constructive suggestion. We explored the cascade reaction with two different aryl halides. As you expected, we only got a mixture of products, because the excess amount of aryl halides need to be added. To solve this issue, we conducted the monoarylation and isolated it, followed by another aryl bromide. You can see we successfully obtained the related product in Fig. 5.

It is not obvious why there is not a second arylation at the ortho C-H bond on the other side of the phosphorus atom. If you look closely at the computations, the energetics suggest that the two C-H insertion reactions are very similar in energy. The authors do not mention if any of this arylation product is formed. Why would it be disfavored? I think this casts a shadow on the quality of the computations if there is high selectivity and the computations put the energy barrier as equivalent. The authors should address in the text if any of the bis-ortho arylation is observed.

Response: Thanks for your questions. The second arylation on a second Ar group of the phosphorus atom was not observed, since this process proceeds via a four-membered chelate ring, which possesses greater ring tension than that process *via* six-membered chelate ring. DFT calculation was also performed to calculate the energy barrier of these transformations. The ortho'-C-H activation undergoes a concerted metalation deprotonation (CMD) process through six-membered cyclic transition state **TS11A-2j** with a free energy of $32.2 \text{ kcal} \cdot \text{mol}^{-1}$, which is much lower than that of four-membered cyclic transition state **TS11B-2j** ($42.8 \text{ kcal} \cdot \text{mol}^{-1}$) and **TS11C-2j** ($42.5 \text{ kcal} \cdot \text{mol}^{-1}$). This part was added in Fig 8c during the revision.

Figure 4: It is unclear to me why the authors included the CD data in this paper. It is a complete non-sequitur. There is no asymmetric reactions that are included and the assignment of the enantiomers is outside of the scope of the manuscript. At most, it could be noted that the two enantiomers of 3aj could be separated by HPLC.

Response: Thanks for this suggestion. We removed the CD data according to your suggestion.

Figure 6b: The aryl halide is mislabeled as 1a. This is rather confusing. I also found the indication of the deuterated substrates as unclear. PPh₃ should be labeled as -d15 in the figure. Furthermore, the indication of d14 on d-3aa is unclear. Each Ph group should be labeled as C6D₅ and D4 could be shown on the aryl group that is drawn out.

Response: We corrected them.

Figure 7: This figure would be much clearer if you replaced 1a with PPh₃. This is a simple fix that would increase clarity significantly.

Response: We changed it.

Line 140: The text says that the calculated TS for TS5A is 31.5. Figure 7 says it is 32.3. Which one is correct? I would not expect a KIE for this reaction if it is not the rate determining step. It is very close to TS4A if the figure is correct, but not if the text is correct. Some comment on the subtle nature of the computations vs. the experiment would be helpful. I also think that a calculation for a deuterated substrate would be ideal to examine the difference.

Response: The original **TS5A** has been revised as **TS4A**. Taking the energy of **INT1A** as zero, the relative energies of **INT3A** and **TS4A** are 0.8 and 32.3 kcal · mol⁻¹, respectively. The energy gap between **INT3A** and **TS4A** is 31.5 kcal · mol⁻¹ (32.3 - 0.8 = 31.5), suggesting that a concerted metalation deprotonation (CMD) process of **INT3A** *via* transition state **TS4A** needs to overcome an energy barrier of 31.5 kcal · mol⁻¹. This process possesses very similar high energy with the C-Br bond oxidative addition (32.3 kcal·mol⁻¹ vs. 32.4 kcal·mol⁻¹ relative to zero point). The theoretical calculations have certain deviations. To further verify whether C-H metalation is the rate-determining step, the deuterated substrate via a deuterated transition state of **TS4A-D** has

been calculated.

The calculated key transition states were shown as follows:

The energy of *ortho*-C-D bond activation is $0.9 \text{ kcal} \cdot \text{mol}^{-1}$ higher than the energy of *ortho*-C-H bond activation. Based on Eyring H equation, we can calculate the theoretical KIE as follows:

$\text{KIE} = k_{\text{H}}/k_{\text{D}} = e^{(0.9 \times 1000)/(1.987 \times 298.15)} = 4.13$, which further proves that *ortho*-C-H bond activation is the rate-determining step.

Figure 8: The authors chose to do the reductive elimination TS for the *t*-Bu substrate and it is over the RDS, but not by a huge amount. I don't think this explains a product like 3ad. I suspect that RE step will be lower than the turnover limiting step. In this case it would be difficult to stop the reaction at 1 arylation if the computations are correct. The authors should show this calculation and comment on the size of that group and how it influences the selectivity. This can be seen in the lower temp and shorter reaction time for 3ad, but no comment is made.

Response: Thanks for your suggestion. We have revised the proposed mechanism. Based on the revised mechanism, when bromoarene **2a** with sterically hindered substituents at the meta positions was used as the substrate, ligand exchange of **INT10A** with **1a** is found to be a favorable process, and its barrier height is $11.2 \text{ kcal} \cdot \text{mol}^{-1}$ lower than the C-H metalation through six-membered cyclic transition state **TS11A** ($18.2 \text{ kcal} \cdot \text{mol}^{-1}$ vs $29.4 \text{ kcal} \cdot \text{mol}^{-1}$), suggesting that one-fold arylation will proceed to the end until **1a** was completely consumed out. In addition, the *meta*-substitutions of bromoarenes increases the difficulty of the second reductive elimination, making the activation barrier of transition state **TS14A** as high as $35.6 \text{ kcal} \cdot \text{mol}^{-1}$. The high energy barrier was mainly attributed to the bulky steric repulsion between the two meta-substituents.

Also in figure 8, it is notable that TS10A and TS10-2j are so similar in energy. Even more notable that TS10a is lower in energy than TS10a-2j, which is less congested. What does this tell you? Some comment on this would be ideal.

Response: The Gibbs free energy change (ΔG) is a relative value, not an absolute value, which depends not only on temperature and pressure, but also on the chemical potentials of the species involved. ΔG cannot be used as a basis for judging the stability of compounds, but $\Delta\Delta G$ is important for determining the likelihood of a reaction. Therefore, the energy difference between two different intermediates from different reactions does not mean anything. The original TS10A and TS10A-2j has been revised as TS14A and TS14A-2j, respectively. The reductive elimination through TS14A need to overcome an energy barrier of $35.6 \text{ kcal} \cdot \text{mol}^{-1}$ (the energy of TS14A $-47.8 \text{ kcal} \cdot \text{mol}^{-1}$ minus the energy of INT13A $-83.4 \text{ kcal} \cdot \text{mol}^{-1}$), while the reductive elimination through TS14A-2j need to overcome an energy barrier of $22.5 \text{ kcal} \cdot \text{mol}^{-1}$ (the energy of TS14A-2j $-69.0 \text{ kcal} \cdot \text{mol}^{-1}$ minus the energy of INT13A-2j $-91.5 \text{ kcal} \cdot \text{mol}^{-1}$). The energy barrier of second reductive elimination through TS14A is much higher than that process through TS10A-2j, this illustrates that second reductive elimination through TS14A-2j is much thermodynamically smoother.

Supporting Information:

The SI is thorough and quite good, but the authors should include the melting points of the solid products.

Response: Thanks for this kind suggestion! The melting points of the solid products have been added in SI according to your suggestion.

Reviewer #4

This work is concerned with a methodology development, reporting a Rh(I)-catalysed protocol that appends arenes to arylphosphines to access a series of biaryl monophosphines via P(III)-directed ortho C-H activation. Experimentally, the results are important and interesting, and well described. Overall, I believe that this work deserves publication in Nature Communications. However, the computational results were rather poorly presented and discussed.

Response: Thanks for such positive comments.

1. In most of the chemdraw structures in the potential energy profiles, the drawings do not reflect the coordination chemistry around the Rh metal center, making readers hard to comprehend the results. We all know that Rh(I) often adopts square planar geometry if it is 4-coordinate and Rh(III) adopts an octahedral if it is 6-coordinate. In all of the drawing, this kind of coordination feature is completely missing.

Response: Thanks for your kind suggestion. The structures of all intermediate and transition state have been redrawn to well reflect the coordination chemistry around the Rh metal center.

2. A potential energy profile for a catalytic cycle should show clearly the “starting” point (Active Catalyst + Reactants) and the “ending” point (The same Active Catalyst + Product). However, readers cannot see this in all the profiles presented in the main text and in the SI.

Response: Thanks for your suggestion, we have revised the catalytic cycle.

3. In the potential energy profiles presented, the ligand exchanges (e.g., INT2A + 2a  INT3a + 1a; INT6a to INT7a; and INT7a + 2a  INT3A + 3aa) are expected to have barriers. The authors do not calculate/include these, which could be high and very important.

Response: The transition states of ligand exchanges have been added in the revised catalytic cycle.

4. Most species are mono-anionic. The authors do not show this in the chemdraw structures presented.

Response: We have added charges to all mono-anionic intermediates.

5. From Fig 7, the Ar-Br OA transition state and the concerted metalation deprotonation (CMD) transition state show almost the same stability. The data in fact show that the OA TS is lying slightly higher than the CMD TS, suggesting that the OA TS is very likely the rate-determining TS. How could the authors claim that the DFT results are consistent with the KIE experimental observation?

Response: Thanks for your suggestion, we have revised this description. The theoretical calculations have certain deviations. In our calculation, the energy of CMD transition state **TS4A** is very close to the C-Br bond oxidative addition transition state **TS3A** (32.3 kcal·mol⁻¹ vs. 32.4 kcal·mol⁻¹). To further verify whether C-H metalation is the rate-determining step, the deuterated substrate via a deuterated transition state of **TS4A-D** has been calculated.

The calculated key transition states were shown as follows:

TS4A-D

$$\Delta\Delta G = 32.5 \text{ kcal mol}^{-1}$$

TS4A-H

$$\Delta\Delta G = 31.6 \text{ kcal mol}^{-1}$$

The energy of *ortho*-C-D bond activation is $0.9 \text{ kcal} \cdot \text{mol}^{-1}$ higher than the energy of *ortho*-C-H bond activation. Based on Eyring H equation, we can calculate the theoretical KIE as follows:
 $\text{KIE} = k_{\text{H}}/k_{\text{D}} = e^{(0.9 \times 1000)/(1.987 \times 298.15)} = 4.13$, which further proves that *ortho*-C-H bond activation is the rate-determining step.

6. In Fig 7, TS4B seems to miss one phosphine ligand.

Response: We have revised the structure of original **TS4B**.

7. It is hard to understand how TS5B transforms to INT5B. This seems to jump too fast, because one more phosphine ligand is included. Then, from INT5B to INT6B, no transition states are involved?

Response: We have revised the catalytic cycle shown in Fig. 7.

8. Did the authors attempt to locate a direct CMD process that converts INT2A to a species containing HOAc as a ligand?

Response: The direct CMD process of original **INT2A** was also calculated. The energy in this pathway through transition states **TS2C** is 2.8 kcal/mol higher than that the C-Br bond oxidative addition through transition state **TS3A**.

9. In Fig 4b, the theoretical ECDs are noticeably different from the experimental ECDs. I do not understand the claim "...were found to match well with the experimental ECD spectra ..."

Response: Thanks for this question. The experimental ECDs are the results of the aggregate of a large number of molecules, and the theoretical ECDs are calculated based on the lower energy conformer of the molecules, thus making the calculated results do not agree well with the experimental results.

Regarding the ECDs, another reviewer also commented: "It is unclear to me why the authors included the CD data in this paper. It is a complete non-sequitur. There is no asymmetric reactions that are included and the assignment of the enantiomers is outside of the scope of the manuscript. At most, it could be noted that the two enantiomers of 3aj could be separated by HPLC."

Therefore, we removed the ECD spectra during the revision.

10. The presentation and discussion on the two- and three-fold arylation reactions are very hard to comprehend. Rewriting of the part is necessary.

Response: This part has been rewritten.

REVIEWER COMMENTS

Reviewer #1 (Remarks to the Author):

The revised manuscript is more promising than the initial version, and most of the issues raised by all reviewers have been properly answered. The part on selectivity is now clear. I recommend its publication.

Reviewer #3 (Remarks to the Author):

The revised manuscript on the C-H arylation of triarylphosphines is much improved from the original submission. The clarity of the manuscript is much better and the confusing aspects were removed. I have a few minor suggestions to further improve the quality and clarity. Ultimately, I am of the opinion that this manuscript is strong and is reasonable for Nature Comm., but is a bit of a stretch for such a high-impact journal. After reading the other reviewer's comments, I agree with Reviewer 2 that the scope is a bit limited, but the work is thorough and can open up directions for the C-H arylation of phosphines. So, I provide a weak endorsement for acceptance in this journal with the following changes:

1) In figure 2, the authors should address if other Lewis bases in place of methoxy work for this reaction. Can a tertiary amine be used? Sulfur? Carbonyl? I assume these were tried and failed. Please mention these results in the text.

2) It is rather peculiar that the 4-membered chelation with triphenylphosphine gives a TS barrier of 31.5 kcal/mol and a similar 4-membered TS after monoarylation has a barrier of 42.8. I did not see an explanation for why this barrier is so much higher. One should be provided.

3) In the SI, the procedure for the KIE has some issues.

a) the procedure says the "indole" was added. No indole is used in this study.

b) The kinetics plot has no units. The plot should be given more detail to provide context for the readers.

Reviewer #4 (Remarks to the Author):

My comments in the previous round of review are mainly related to some technical issues. The authors have satisfactorily addressed the questions I had.

Reviewer #3 (Remarks to the Author):

The revised manuscript on the C-H arylation of triarylphosphines is much improved from the original submission. The clarity of the manuscript is much better and the confusing aspects were removed. I have a few minor suggestions to further improve the quality and clarity. Ultimately, I am of the opinion that this manuscript is strong and is reasonable for Nature Comm., but is a bit of a stretch for such a high-impact journal. After reading the other reviewer's comments, I agree with Reviewer 2 that the scope is a bit limited, but the work is thorough and can open up directions for the C-H arylation of phosphines. So, I provide a weak endorsement for acceptance in this journal with the following changes:

1) In figure 2, the authors should address if other Lewis bases in place of methoxy work for this reaction. Can a tertiary amine be used? Sulfur? Carbonyl? I assume these were tried and failed. Please mention these results in the text.

Response: Thanks for this kind suggestions. Indeed, the selection of OMe at *ortho* position of arylbromides had a strong impact on the reactivity, other functional groups including NMe₂, SMe and Ac led to very low conversions under the current reaction conditions. We added the related information in the text.

2) It is rather peculiar that the 4-membered chelation with triphenylphosphine gives a TS barrier of 31.5 kcal/mol and a similar 4-membered TS after monoarylation has a barrier of 42.8. I did not see an explanation for why this barrier is so much higher. One should be provided.

Response: Thanks for this kind suggestion. After monoarylation, dihedral angle of biphenyl is nearly 90° in intermediate **INT10A-2j**, which is a stable conformer with lower energy. In the disfavored transition states **TS11B-2j** and **TS11C-2j**, the dihedral angle of biphenyl is deformed to 75.1° and 64.3°, respectively, indicating the distinct repulsion between the incorporated aryl group and second aryl group of P, thus increasing the activation energy barriers to > 40 kcal/mol. The explanation has been added in the revised manuscript.

3) In the SI, the procedure for the KIE has some issues.

a) the procedure says the "indole" was added. No indole is used in this study.

Response: We corrected it.

b) The kinetics plot has no units. The plot should be given more detail to provide context for the readers.

Response: Thanks for this kind suggestion. Units have been added in the kinetics plot. And we have marked the yields for each point on the chart.